# A Feasible Compatibilization Processing Technique for Improving the Mechanical and Thermal Performance of Rubbery Biopolymer/Graphene Nanocomposites

**DOI:** 10.3390/polym14225009

**Published:** 2022-11-18

**Authors:** Dalila Shahdan, Noor Afizah Rosli, Ruey Shan Chen, Sahrim Ahmad

**Affiliations:** 1Department of Applied Physics, Faculty of Science and Technology, Universiti Kebangsaan Malaysia, Bangi 43600, Selangor, Malaysia; 2Department of Chemical Sciences, Faculty of Science and Technology, Universiti Kebangsaan Malaysia, Bangi 43600, Selangor, Malaysia

**Keywords:** processing technique, compatibilizing, polymer matrix nanocomposite, tensile property, thermal stability

## Abstract

Over the last few decades, processing and compatibility have become challenging and interesting investigation areas of polymer matrix nanocomposites. This study investigated the addition of maleic anhydride (MAH) at different ratios with graphene nanoplatelets (GnPs) in poly(lactic acid)/modified natural rubber/polyaniline/GnP (PLA/m-NR/PANI/GnP) nanocomposites via two processing methods: a two-step technique and a one-pot technique. The former technique involved first preparing a master batch of PLA grafted with MAH, followed by a second step involving the melt blending of the nanocomposite (T1) using MAH-g-PLA. On the other hand, the one-pot technique involved the direct mixing of MAH during the melt-blending process (T2). The mechanical, morphological and thermal properties of the prepared nanocomposites were investigated. The findings showed that adding MAH significantly improved the tensile strength and elongation at break by about 25% for PLA/m-NR/PANi/GnP nanocomposites, with an optimal ratio of 1:1 of MAH-g-PLA to GnP loading using the T1 technique. FTIR analysis confirmed the chemical interaction between MAH and PLA for T1 nanocomposites, which exhibited improved phase morphology with smoother surfaces. MAH-compatibilized nanocomposites had enhanced thermal stabilities when compared to the sample without a compatibilizer. The findings show that the compatibilized PLA nanocomposite is potentially suitable for bio-inspired materials.

## 1. Introduction

Studies on polymer matrix nanocomposites have been widely conducted, as they can promote the best performance with versatile properties [1,2,3,4,5,6] and have potential applications in many fields, such as biomedicine [7,8] and electronic devices [9], and even in daily use, such as food packaging [10]. Despite that, the most debated issue is that polymeric base materials lead to solid waste problems since they only degrade or decompose after a very long time. Due to this, studies on green products and environmentally friendly materials have been intensively conducted in recent years. Environmentally friendly aliphatic polyester families, such as poly(lactic acid) (PLA), are claimed to be the best green product candidates, as they can be synthesized from renewable agricultural products such as corn starch and sugarcane. PLA exhibits high mechanical strength, non-toxicity, transparency, bio-compatibility and bio-degradability. Nevertheless, the brittleness property of PLA has been a major hindrance to its use in many applications and industries. In order to improve the toughness (ductility) properties of individual PLA, research on PLA blending with modified liquid natural rubber (m-NR)/polymers [11,12] and fillers/reinforcements such as carbon nanotubes (CNTs) [13], nickel–zinc (NiZn) ferrite [14] and cellulose [15] has been performed. Our previous study proved that the inclusion of polyaniline (PANi) in a PLA/rubber blend enhanced the mechanical and thermal properties in addition to promoting the conductive properties, even at minimal amounts (ranging from 0.03 to 0.11 wt%) [5,16]. 

In addition to single-filler reinforcement, the hybridization concept is established to be an effective way to provide synergistic effects on the resultant performance. For instance, combining carbonaceous (i.e., graphene nanoplatelets (GnPs)) and pseudocapacitive (PANi) materials was shown to improve mechanical flexibility and, at the same time, enhance the electrode properties of the materials [9]. However, nano-sized fillers have a high tendency to agglomerate in the polymer matrix [13]. In this sense, our previous studies applied an ultrasonic treatment to assist in the mechanical exfoliation and dispersion of particles in the nanocomposite. The findings of studies on a NiZn ferrite nanofiller alone, PANi alone and hybrid GnP/PANi-based PLA showed that 1 h (which was used in this study) ultrasonication was optimal for the enhancement of mechanical (~23% and 117% increments in tensile strength and elongation at break in the latter study) and thermal properties [5,17]. These findings showed that ultrasonic treatment majorly contributed to the improvement in terms of the dispersion and distribution of the reinforcing particles [5,17]; however, they did not provide any information on the chemical or physical interaction between the components in the composites. 

The literature reveals that the compatibility between the polymer matrix and nanofiller can be a critical issue influencing interfacial adhesion and bonding, which subsequently affect the final performance. To solve this problem, the inclusion of a compatibilizer could be effective since it may result in a better and more stable morphology and the improved adhesion of the polymer blend [18]. Maleic anhydride (MAH) is the most commonly used reagent for functionalizing the PLA chain because of its excellent chemical reactivity toward PLA free radicals induced by an initiator, low toxicity and good chemical stability, as well as a low potential to polymerize itself under free radical grafting conditions [19,20]. The succinic anhydride groups of MAH-grafted polymers are highly reactive, and covalent bonds can be formed with polar polymer backbones and end groups [21]. Several reports have investigated the grafting process of PLA and MAH in blends or nanocomposites with different routes: (1) with organic peroxides (e.g., benzoyl peroxide (BPO) [22] or dicumyl peroxide (DCP) [19,23]) as initiators via a two-step method, where MAH-g-PLA was first prepared before its use in melt compounding, (2) without an initiator via a two-step method [24], and (3) with and without organic peroxides via a one-pot technique [25]. With MAH-g-MA, the interfacial adhesion and miscibility of PLA/poly(butylene sebacate-co-terephthalate [23] and PLA/starch [19] blends were increased, as confirmed by the improvement in tensile strength, elongation and thermal stability. The maleation effect was also proven to be efficient for nanocomposite systems with clay nanoparticles, where an extraordinarily well-dispersed structure was obtained through a masterbatch module [24], and carbon nanotubes had a better distribution in the matrix [22]. 

Although much research has been conducted on compatibilized PLA, there have been very few studies related to the comparison of grafting methods specifically on m-NR-toughened PLA polymer nanocomposites, where m-NR is believed to further facilitate the effectiveness of compatibilization due to the presence of C=O, OH and -OOH [11]. In this study, the aim was to investigate the mixing method, either the two-step or one-pot technique, used for the MAH grafting purpose in order to improve the chemical interaction and the dispersion/distribution of GnPs in the PLA-based matrix. Deep insight into the grafting mechanism in the PLA/MAH/DCP system was revealed based on spectroscopic analysis. Another emphasis of this study was the investigation of the effect of the compatibilizer-to-nanofiller ratio on the mechanical and thermal stability.

## 2. Experiments

### 2.1. Materials

Polymer matrix: PLA (3251D) was purchased from NatureWorks^®^ Ingeo Biopolymer. MAH and DCP with a pellet shape with a purity of 99% and 98%, respectively, were purchased from Sigma Aldrich. PANi (PA-N35S; emeraldine base), with a dark green color and a needle-like shape, was purchased from E-TEK Co. Ltd., Korea. Self-synthesized modified liquid natural rubber (m-NR) (Grade: SMR-L) with 41% of its total solid content was prepared via a photosensitized chemical degradation/oxidation technique [5]. The properties of raw materials in the nanocomposites used in this study are described in Table 1.

Nanofiller: The GnP (Type: KNG-150) used in this study was distributed by Xiamen Knano Graphene Technology Corporation. It has an average thickness of ~15 nm with a carbon content of >98 wt%.

### 2.2. Nanocomposite Sample Preparation

PLA/m-NR/PANI/GnP nanocomposites were prepared by two different techniques: Technique 1 (T1) and Technique 2 (T2), as shown in Figure 1. T1 was a two-step technique, which consisted of two steps, in which the process began by preparing MAH-g-PLA via the melt-blending method and using it as a compatibilizer with the PLA/m-NR/PANi/GnP nanocomposites. In the first stage, MAH-g-PLA was prepared using a HAAKE Rheomix OS Lab Mixer (Thermo Fisher Scientific, Waltham, MA, USA). PLA was first discharged into the internal mixer and left to melt for 3 min, and then 3.6 phr DCP and 9 phr MAH were added after 3 min and 6 min, respectively. The second stage of melt blending was used to prepare the nanocomposites via the same mixer at 190 °C and 100 rpm for 15 min. A constant ratio of PLA to m-NR (90:10 wt.%) was maintained in each formulation. m-NR was pre-mixed with PANi (0.09 wt.%) and GnPs (0.4 wt.%) using an ultrasonic bath for 1 h. PLA was first added to the internal mixer and left to melt for 3 min. Then, MAH-g-PLA was added with different ratios of compatibilizer to filler (1:1, 1:2 and 2:1 wt.%). Lastly, the pre-mixed m-NR/PANi/GnPs were introduced into the internal mixer.

Meanwhile, T2 involved the reactive compatibilization of MAH via a “one-pot” technique to make the nanocomposites by directly combining PLA, m-NR, PANi, GnPs and MAH with the DCP initiator. All processing parameters and steps were fixed as in T1. The control sample without the PLA/m-NR/PANi/GnP compatibilizer, namely, NC, was prepared via melt blending using the processing parameters used in the T1 technique.

The mixed materials were then cut into small sizes, and then they proceeded to hot and cold pressing and were molded into the shape according to the tested standard. Hot pressing was performed at a temperature of 190 °C for 13 min (4 min preheating, 4 min venting and 5 min full pressing), while cold pressing was performed for 3 min.

### 2.3. Characterization 

The tensile properties of the nanocomposites were studied using a tabletop universal tensile machine (Testometric M350-10CT, The Testometric Company Ltd., Rochdale, UK) with a 5 kN load cell and 5 mm/min cross-head speed. The testing system was run according to ASTM D638-03, Type-I. Six specimens were tested to determine the average value. The tensile-fractured surface morphologies of graphene nanocomposites with and without a compatibilizer were examined via variable-pressure SEM (VPSEM), model Philips XL-30 (F.E.I. Company, Hillsboro, OR, USA). The samples were coated with a thin layer of gold to avoid any electrostatic. The morphologies of the nanocomposites were observed at a 13 kV accelerating voltage with 500× magnification.

In order to determine the functional groups and any chemical reactions that occurred in the T1 and T2 samples, an attenuated total reflectance infrared (ATR-IR) analysis was performed. The ATR-IR spectra were recorded using a Cary630 Agilent Technologies spectrometer (Agilent Technologies, Inc., Santa Clara, CA, USA). With a scanning resolution of 4 cm^−1^, the wavenumber range used in the study was between 4000 and 600 cm^−1^. An average of 16 scans was used in order to obtain each spectrum.

The thermal stabilities of the nanocomposites were determined using thermogravimetric analysis (TGA) and a differential scanning calorimeter (DSC) (Mettler Toledo models: TGA/SDTA851^e^ and DSC 882^e^, respectively). Approximately 10–15 mg of the sample was tested at a heating rate of 10 °C/min under an atmospheric nitrogen gas flow rate of 10 mL/min condition. The temperature range was fixed from 25 to 600 °C for TGA and from 25 to 250 °C for DSC. For DSC analysis, a second scan was performed to record the heat effect of the nanocomposite system and also to avoid the effect of the heat history. The temperature peaks obtained were the glass-transition temperature (T_g_), crystallization temperature (T_c_) and melting temperature (T_m_). The degree of crystallinity (χ_c_) of the PLA nanocomposite system in the second scanning curve of its melting behavior was evaluated according to Equation (1) with the incorporation of Origin 2019b (version 9.65) software:(1)$$\chi_{\mathrm{cDSC}}=\frac{\Delta H_{m}}{\Delta H^{0}{}_{m}\times \Phi_{\mathrm{PLA}}}\times 100\%$$
in which $\Delta H_{m}$ represents the heat fusion of the sample; $\Delta H^{0}{}_{m}$ represents the heat of fusion for 100% crystalline PLA, and $\Phi_{\mathrm{PLA}}$ represents the net weight fraction of PLA. The heat fusion of 100% crystalline PLA is 93.7 J/g [26].

The thermal stability was further studied via integral procedure decomposition temperature (IPDT) calculation. The area under the TGA curve was measured using Origin 2019b (version 9.65) graphing and analysis software. The IPDT was evaluated [27] as follows:IPDT (°C) = *A* K** (*T_f_* − *T_i_*) + *T_i_*(2)
*A** = (*S*_1_ + *S*_2_)/(*S*_1_ + *S*_2_ + *S*_3_)(3)
*K** = (*S*_1_ + *S*_2_)/*S*_1_(4)
where *A** represents the area ratio of the total TGA experimental curve with respect to the experimental temperature range, *K** represents the coefficient of *A**, and *T_i_* and *T_f_* represent the initial and final experimental temperatures used for the TGA test, respectively. Meanwhile, *S*_1_, *S*_2_ and *S*_3_ for *A** and *K** calculations represent the area in the experimental TGA curve, as shown in Figure 2.

The intercalation and exfoliation states of the nanofiller in the polymer matrix were studied using X-ray diffraction (XRD) analysis by using a D8 Advance diffractometer with CuKα radiation with an operating voltage of 40 kV and a current of 30 mA. Angles ranging from 5° to 30° were used to scan the samples. The crystallinity of the nanocomposite system was evaluated based on the comparison of the area under the XRD peaks. The area under the XRD peaks was generated using Origin 2019b (9.65), and the degree of crystallinity can be written as Equation (5) [28]:(5)$$C_{r}=\frac{Area\;of\;crystalline\;fraction\;\left(under\;peak\right)}{Area\;of\;crystalline\;fraction+Area\;of\;amorphous\;fraction}$$

Hence, the crystallinity percentage of the nanocomposite system was evaluated according to Equation (6) as follows:(6)$$\chi_{\mathrm{cXRD}}=\frac{Area\;of\;crystalline\;fraction\;\left(under\;peak\right)}{Area\;of\;crystalline\;fraction+Area\;of\;amorphous\;fraction}\times 100$$

## 3. Results and Discussion 

### 3.1. Tensile Test

Figure 3 shows the effect of the MAH compatibilizer’s presence, its composition ratio and the mixing method on the tensile properties of the samples. In general, the effect of different compatibilizer/GnP ratios (1:1, 1:2 and 2:1) was different between techniques T1 and T2. By comparing the mixing techniques, the overall results showed that samples prepared via T1 exhibited a positive effect as compared to T2. This is because with the one-pot technique (T2), the existence of free radicals from the initiator contributed to the degradation of the whole nanocomposite system; hence, low tensile properties were obtained [29]. Specifically, the sample with a compatibilizer prepared via T1 with a 1:1 ratio of compatibilizer to GnPs showed the best tensile properties, having a 26.8 MPa tensile strength, a 1229.8 MPa tensile modulus and 5.0% elongation at break; meanwhile, T2 samples showed no improvement and had lower tensile results than NC (control sample). 

By looking at the optimal improvements in the tensile strength (by 8.2%), tensile modulus (by 3.4%) and elongation at break (by 31.4%) for a ratio of 1:1 (T1), it can be observed that this ratio (1:1) provided the best combination among the polymers and nanofiller. The addition of MAH-g-PLA in the appropriate amount is believed to optimally improve the interfacial adhesion between nanoplatelets and the polymer matrix, as evidenced by the SEM micrograph in Figure 4b, where the dispersion of the nanofiller is more homogeneous compared to NC and T2 nanocomposites (Figure 4a,c, respectively). The improved interfacial adhesion could allow effective stress transfer to take place between the nanofillers and the polymer matrix, thereby enhancing the mechanical properties of the compatibilized nanocomposite. This finding is in agreement with Nyambo et al. [29], who reported that adding 3 phr PLA-g-MAH (8:2 ratio of MAH to PLA) could improve the interfacial adhesion due to covalent and hydrogen bond formation between the polymer matrix and filler loading, and hence, the stress could be transferred easily. In addition to the tensile strength and modulus, the elongation at break of this sample (nanocomposites compatibilized with MAH/GnPs (1:1) and prepared by T1) was seen to be greatly improved. This is because of the capability of the MAH-g-PLA compatibilizer to induce an energy dissipation mechanism in the PLA nanocomposite system during tensile deformation. Therefore, the polymer matrix will absorb the energy and prevent a highly localized strain process, thus increasing the elongation at break [30]. 

When using a lower content of the MAH compatibilizer (which was a 1:2 ratio of MAH-g-PLA to GnPs), the tensile strength of T1 nanocomposite (24.4 MPa) samples was not much different from that of the samples without the compatibilizer (24.8 MPa). This is probably due to the insufficient content of the compatibilizer for a compatibilizing effect on the resultant nanocomposites. Meanwhile, at a MAH-g-PLA/GnP ratio of 2:1, the tensile properties were found to deteriorate as compared to the 1:1 ratio and even NC samples. The trend obtained can be explained by the presence of anhydride groups in the nanocomposite system, the number of acid–base interactions and the degree of hydrogen bonding, leading to stronger interactions between the filler and the matrix. However, excess amounts of anhydride groups could lead to a counteracting effect due to the hydrolysis of the bulk polymer, hence reducing the tensile properties of the nanocomposites.

On the other hand, as shown in the plotted graph in Figure 3, the tensile properties of T2 samples showed no significant trend with different ratios of compatibilizer to GnPs. They also cannot compete with T1 samples because the interactions in T1 samples are relatively strong between MAH and the filler with the use of polymer-grafted MAH [25]. Through the reactive compatibilization of T2 samples, the existence of free radicals from the initiator causes the degradation of the polymer matrix and, thus, causes a drop in tensile properties [25]. In addition, the insufficient duration or time of melt blending during the one-step technique (T2) could be another reason for the lower tensile properties compared to the two-step technique. As reported by Shi et al. [25], an optimal duration of 7 min was needed to allow the optimal reaction of the compatibilizer; however, in this study, the duration was less than 7 min (6 min). 

### 3.2. Morphological Analysis

Figure 4a–c represents tensile specimens before and after the tensile test. All samples experienced fracture without elastic deformation (elongation effect). It is observed that the fractured surface, as highlighted by the red dashed-line box in Figure 4b for the T1 sample, has more of an elongation effect, as shown by the formation of the white area on the surface, than that in Figure 4a,c. This indicates the T1 sample has higher strength and elongation at break compared to the NC and T2 samples. Figure 4 demonstrates SEM micrographs at 500× and 1000× magnification of (a’) NC, (b’) T1 and (c’) T2 with a ratio of MAH/GnP of 1:1, which showed the best tensile performance. In Figure 4a’, it can be observed that many holes or voids were present in the NC sample, which represents the existence of m-NR. In the presence of the MAH compatibilizer, the results showed that the number of pores was reduced, and the formation of a stronger stretching effect can be seen on the tensile-fractured surface (Figure 4b’,c’), irrespective of the mixing method. As highlighted by the red circle in Figure 4b’, fewer and smaller holes in the matrix indicate the homogeneous dispersion of the GnP nanofiller in the matrix. On the other hand, the observation of a greater stretching effect, as indicated by the red circle in Figure 4b’, suggests good adhesion between the GnP nanofiller and the PLA/m-NR/PANi polymer matrix, which allows a better load transfer between the filler and matrix [31]. The improved adhesion may be due to the coupling of the anhydride groups in MAH-g-PLA with the hydroxyl groups on the surface of PANi or nanofiller GnPs via hydrogen bonding, hence the reason why the tensile properties were higher for the T1 sample. 

Comparing the mixing methods (Figure 4b’,c’) shows that the T2 sample also exhibited a reduced number and size of pores as well as a strong stretching effect on its morphological surface. However, as can be seen from the morphological structure in Figure 4c’, some small and big bright dots are present, which represent the agglomeration of the nanofillers. This indicates the ineffectiveness of the T2 method for mixing the nanocomposite components.

### 3.3. Spectroscopic Analysis

Figure 5 shows the differences in the ATR-IR spectra of the resulting nanocomposites produced by various techniques. Due to the high amounts of PLA and LNR in the nanocomposite system, most observed absorption peaks indicate these materials. There is a noticeable difference between the three spectra when compared at wavenumbers of 3400 cm^−1^ (T2), 1748 cm^−1^ (T1) and 1545 cm^−1^ (T2).

The T1 technique for producing the nanocomposite produced was first started by preparing a large batch of PLA-g-MAH. After PLA-g-MAH was prepared, it was combined with the matrix and nanofillers of PLA, LNR, PANI and GnPs. The ATR-IR spectrum of the T1 nanocomposite showed a new absorption peak at 1756 cm^−1^, indicating the C=O of MAH, thus verifying the production of PLA-g-MAH. However, it was not formed in large amounts due to the low reactivity of MA toward macroradicals due to the low content (0.4 wt.%) of MAH-g-PLA in the nanocomposite compound. A similar trend can be found in a previous study by Ma et al. [30] on the efficiency of MAH grafting onto PLA chains. In addition, the absence of a peak at 1560 cm^−1^, which represents the cyclic C=C stretching of MAH, also shows the success of PLA-g-MAH production. This is because, as illustrated in Figure 1a, the process of grafting MAH onto PLA takes place at the cyclic C=C of MAH. The first step involved in the mechanism for the MAH grafting of PLA via T1 started with the production of primary radicals from DCP decomposition, which later initiates macroradicals of PLA by the abstraction of hydrogen. Afterwards, these PLA macroradicals react with MA to produce PLA-g-MAH, as shown in Figure 1a. The second step in the T1 technique involves mixing PLA-g-MAH with other materials (PLA, LNR, PANI and GnPs) to produce the complete nanocomposite system. The decrease in peak intensity at 1750 and 1360 cm^−1^ (Figure 5), representing C=O and C-O-C stretching vibrations in PLA, respectively, also indicates that a chemical interaction occurred between the components of the nanocomposite, as shown in Figure 1b.

Meanwhile, the nanocomposite spectrum using the T2 method shows an absorption peak with the same intensity as in the nanocomposite spectrum without MAH and DCP, except for the presence of new absorption peaks at 3400 cm^−1^ and 1560 cm^−1^. The presence of an absorption peak at 1560 cm^−1^ indicates that the grafting of MAH onto PLA was less effective because the grafting site occurs at the cyclic C=C of MAH, as illustrated in Figure 1a. This happens because the simultaneous blending of DCP and MAH with all nanocomposite components may cause the formation of radicals on the PLA chain and on other polymeric materials. This, in turn, causes a decrease in the effectiveness of MAH grafting onto PLA. In addition, the presence of a new broad peak at 3400 cm^−1^, which represents the OH group, may also be caused by the combination of peroxide radicals (RO•) with hydrogen (H) from the chain of various polymeric materials in the nanocomposite, which produces high ROH in the nanocomposite system. In addition, the broad peak of OH may be due to the hydrolysis reaction that occurred on the MAH monomer, as shown in Figure 2. The presence of little water due to the highly hydrophilic nature of PANI may cause the opening of the anhydride ring of MAH to produce two carboxylic acids at once, causing an increase in the content of OH groups in the T2 method nanocomposite system. The enhanced and reduced tensile characteristics of nanocomposites using the T1 and T2 techniques are explained and supported by the overall ATR-FTIR data.

### 3.4. Thermogravimetric Analysis (TGA)

The thermal decomposition analysis on the compatibilized and non-compatibilized PLA-based nanocomposite prepared with different mixing methods is presented in Figure 6. As shown by the inset graph in Figure 6a, all of the nanocomposite systems started to become thermally unstable in the temperature range (T_5%_) of 310–328 °C, where T_5%_ can be defined as the temperature at which the weight loss reached 5%. This 5% weight loss started with the T2 sample (the earliest occurrence of 5% weight loss), followed by T1 and, lastly, NC (the latest occurrence of 5% weight loss) nanocomposites. A similar result was found in a previous study by Ma et al., who grafted MAH onto PLA [32]. Usually, these small losses of the sample weight would not affect or contribute to the major degradation reaction or characteristics, as it may be due to absorbed water or moisture weight loss [33]. It might also be due to the earlier decomposition of the compatibilizer [34]. However, at a rapid decomposition rate, where a weight loss of approximately 50% of the nanocomposites (T_50%_) was recorded at 360–368 °C, PLA without MAH (NC sample) was the fastest, as visible from the maximum DTG peak in Figure 6b, when compared to compatibilized nanocomposites (T1 and T2). This shows that even though MAH addition can lead to early decomposition [32], it also needs a high temperature to be decomposed due to the different degradation mechanisms of polymers with and without MAH.

Referring to the derivative curve in Figure 6b, the major weight loss in the TGA curve (in Figure 6a) indicates the clear transformation of the nanocomposite system, which started at 271 °C and ended at a temperature of 435 °C. The relation between the TGA and DTG curves is important to note because it is hard to identify the exact temperature where phase transformation takes place in TGA. However, in the DTG curve, phase transformation can be easily observed at the exact temperature value in the form of an endothermic peak for each transformation. The DTG peaks of the compatibilized nanocomposite system were placed at higher temperatures (361.3 °C for T1 and 364.0 °C for T2 as compared to 359.0 °C for NC), which indicates that the MAH component conveyed definite effects on the thermal stability of the PLA nanocomposites. For the residual amount, the results showed that T1 had the highest amount with 6.5%, followed by T2 with 5.1% and, finally, NC with 3.9% at 550 °C. This phenomenon implies that the addition of a compatibilizer can promote the interaction between the degradation process of matrix components and nanofillers.

Numerically, the level of thermal stability improvement of the nanocomposites can also be determined using the IPDT calculation. In this regard, Doyle [27] came up with the idea of correlating the volatile parts of polymeric materials and used it to estimate the inherent thermal stability of polymeric materials. By applying the IPDT calculation, the evaluation takes into account the whole TGA curve shape in a single number by measuring its area under the curve. From the IPDT results in Table 2, the calculated IPDT of the uncompatibilized nanocomposite was 482.5 °C. On the other hand, with MAH addition, the IPDT values for T1 and T2 samples were higher with increments of 9–15 °C, proving that compatibilization is able to enhance the thermal stability of the nanocomposite system due to the improved polymer–nanofiller interfacial interaction. These findings are in agreement with a previous study on the effect of a compatibilizer on PLA/NR blends [35]. Aligning with the DTG curves (Figure 6b), the T2 sample exhibited slightly better thermal stability, as indicated by the higher IPDT value of 497.7 °C, when compared to the T1 sample (492.2 °C). This can be explained by the fact that the greater interaction in the nanocomposite system (in this case, T1) may reduce the decomposition temperature [36].

### 3.5. Differential Scanning Calorimetry (DSC)

Figure 7 shows the DSC diffractograms of all investigated samples with the labeling of each glass-transition temperature (T_g_), crystallization temperature (T_c_) and melting temperature (T_m_). When comparing T_g_, it is observed that MAH-compatibilized nanocomposites (both T1 and T2) had lower T_g_ (59.9 °C) than that of the uncompatibilized sample (61.0 °C). This finding was found to be similar to that of Muenprasat et al. [25], who also obtained lower T_g_ when MAH was added to PLA. This indicates that samples modified with MAH were less stiff than the unmodified PLA nanocomposite. During heating, the PLA-based samples were expected to crystallize, as indicated by the exothermic peak of the DSC thermogram, in which the crystallization started at 96 °C and ended at around 140 °C. The NC sample exhibited T_c_ at 112 °C, and this value decreased slightly (which was not a significant change) to 111.6 °C and 111 °C for T1 and T2, respectively. Upon further heating, the process reached the melting process, which was shown by the endothermic peak at around T_m_ ~169 °C. Similar to T_c_, the T_m_ peaks of samples with the compatibilizer (T1 and T2) also showed left shifts to a lower temperature (~167 °C), with a negligible difference between the two samples. The insignificant changes in either T_g_ or T_c_ and T_m_ peaks are supported by a previous study on PLA/PLA-grafted MAH/talc composites, which proved similar trends in both T_g_ and T_m_ peaks [26]. However, the T2 sample showed a further drop in T_c_ and T_m_ as compared to the T1 sample and an even greater difference from the NC sample. The reason for this might be the poor blending of MAH in PLA during the mixing process of the T2 method [18]. In addition, the existence of a shoulder peak at the melting temperature (T’_m_) of the T2 sample was more visible compared to T1. The appearance of T’_m_ in T2 could be due to recrystallization, leading to a rise in T’_m_ after the partial melting process. Additionally, the shoulder peak may correspond to nucleation behavior in the crystallization phase of PLA, which results in the rearrangement of polymer lamellae and the reorganization of less crystalline regions in the crystalline structure of PLA [37].

As listed in Table 3, the crystallinity percentage of the samples calculated based on DSC (χ_cDSC_) showed that the T1 and T2 samples had lower values compared to the NC sample. Interestingly, this trend was similar to χ_c_, which was obtained from the XRD results. Compared to NC and T2, T1 had the lowest crystallinity. This can be attributed to the formation of a random interface [38]. A similar trend was observed by Kuila et al. [38], who stated that the decreasing trend of crystallinity was due to the presence of homogeneously distributed dodecyl amine-modified graphene layers in the linear low-density polyethylene matrix, which inhibited the ordered crystalline structure of the polymer chains. In our study, the good interaction of MAH-g-PLA in the nanocomposite blend produced a homogeneous distribution of GnPs in the polymer matrix that led to decreased crystallinity.

### 3.6. XRD Analysis

Figure 8 shows the XRD diffractogram of the three investigated nanocomposite samples. From these observations, the sample with MAH-g-PLA prepared via the two-step technique (T1) showed a similar pattern to NC and T2, but with lower intensity. The peak at 2θ~16°–17°, which represents the semi-crystalline characteristic of the polymer matrix PLA/m-NR/PANi, shifted slightly to the left for the NC and T2 samples as compared to the T1 sample, with a decrease in peak intensity at 2θ = 16.47°. This is due to the charge transfer interactions between PANi and GnPs, leading to variations in chain packing and configurations. A similar trend can also be seen in a previous study [39]. The decreased intensity of XRD peaks can be correlated with the crystallinity results in Table 3, where T1 and T2 samples show that the reduced crystallinity percentage in T1 was the lowest. This is because the grafting of MAH in PLA increased the expansion of amorphous regions. As mentioned by Wang et al. [40], the scattering intensity in the SAXS profile of polypropylene decreases due to the lower density difference between neighboring PP crystal lamellae and amorphous lamellae because of the increased density in the amorphous region with MAH addition.

As established, nanocomposites containing GnPs typically showed a graphitic characteristic at 2θ = 26.6° [41]. However, this peak did not appear in the T1 sample, suggesting fully exfoliated GnPs in this nanocomposite with the aid of MAH-g-PLA due to the strong interfacial adhesion obtained. This phenomenon greatly supports the earlier result for tensile strength. On the other hand, the NC and T2 nanocomposite samples showed different diffractograms, as highlighted at 2θ = 28.7° (in Figure 8). Batakliev et al. [13] stated that the position change indicates that the spacing between GnPs in the nanocomposites is shortened. In this case, it is believed that these results indicate an increased interlayer spacing of GnPs, which may correspond to the intercalation of GnPs.

## 4. Conclusions

In this research, modified polymeric rubber (PLA/m-NR/PANi/GnP) nanocomposites were successfully prepared by adding MAH as a compatibilizer via a two-step technique that started by preparing a masterbatch of PLA-g-MAH, which was then used as a compatibilizer in the resultant nanocomposites. Based on tensile properties, the results showed that this two-step technique was better than the one-pot technique, which involved direct mixing in an internal mixer with the aid of DCP. A MAH-g-PLA-to-GnP ratio of 1:1 in the nanocomposite was found to be the best ratio to give the optimal tensile performance, with an 8.2% and 31.4% increase in its tensile strength and elongation, respectively. The SEM micrograph of this compatibilized sample proved the improved interaction and interfacial adhesion with the fully exfoliated GnPs in the polymer matrix, as evidenced by the disappearance of the graphitic characteristic in the XRD results. TGA curves that shifted to the right to a higher temperature for compatibilized nanocomposites proved their enhanced thermal stability. From these findings, it can be concluded that these modified polymers–natural rubber nanocomposites have the potential to be used as a green alternative material. Further specific characterizations, such as electrical conductivity and biocompatibility, can be investigated for various applications, especially in the medical field.

## Data Availability

Not applicable.

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
