# Peer review of "A Feasible Compatibilization Processing Technique for Improving the Mechanical and Thermal Performance of Rubbery Biopolymer/Graphene Nanocomposites"

_polymers, 2022, doi:10.3390/polym14225009_

Round 1
Reviewer 1 Report
see attachment

Author Response
Response to Reviewer 1 Comments
Point 1: Singular/plural problem in Line 37-38.
Response 1: The highlighted sentence has been revised as shown in Line 38-39, with some changes in the content based on the comment from Reviewer 3.
Point 2: Please name the T-Scale (°C).
Response 2: The T-Scale unit (°C) has been added as shown in Table 2.

Reviewer 2 Report
This paper presents an analysis on the characteristics of rubbery biopolymer/graphene nanocomposites. The paper may be of interest to the scientific community through the topic addressed. Authors should consider the following observations:
- The Introduction section needs to be improved, because very few articles from the last 5 years were considered. Also, at the end of the Introduction, the structure of the paper and the objectives of the research should be detailed more clearly;
- It is necessary to present the physical-mechanical properties of the materials used in the research;
- It is strictly necessary to present macroscopic images of the samples, both before and after the tests performed;
- The SEM images must have a better resolution and, at the same time, a complete analysis of them must be carried out;
- Conclusions should be more concrete and future research directions should be presented.
Author Response
Response to Reviewer 2 Comments
Point 1: The Introduction section needs to be improved, because very few articles from the last 5 years were considered. Also, at the end of the Introduction, the structure of the paper and the objectives of the research should be detailed more clearly;
Response 1: The whole introduction section has been intensively improved, in terms of flow, structure, state-of-art contents and the cited references. The objectives of the research have been revised in Lines 109-115: “In this study, the aim was to investigate the mixing method, ….Another emphasis of this study was the investigation of compatibilizer to nanofiller ratio on the mechanical and thermal stability analysis”
Point 2: It is necessary to present the physical-mechanical properties of the materials used in the research
Response 2: The physical properties of the material have been included in materials section 2.1 (125-126) and Table 1.
Point 3: It is strictly necessary to present macroscopic images of the samples, both before and after the tests performed;
Response 3: The macroscopic images of the tensile specimens have been included in Figure 4 (a)-(c), with the relevant discussion in Line 275-280.
Point 4: The SEM images must have a better resolution and, at the same time, a complete analysis of them must be carried out;
Response 4: A better resolution of SEM has been provided in Figure 4 (a’-c’) with an additional focus area with 1000x magnification of each specimen. Discussion has been revised in Line 275-282.
Point 5: Conclusions should be more concrete and future research directions should be presented.
Response 5: The conclusion has been improved to be more concrete and future research directions have been suggested in Lines 495- 503: “…SEM micrograph of this compatibilized sample proved the improved interaction and interfacial adhesion with the fully exfoliated GnP in the polymer matrix as evidenced in XRD result by the disappearance of graphitic characteristic. The TGA curves that shift-ed to the right to a higher temperature for compatibilized nanocomposites proved the enhanced thermal stability. From these findings, it can be concluded that these modified polymer-natural rubber nanocomposites are the potential to be used as a green alternative material. Further specific characterizations such as electrical conductivity and biocompatibility can be investigated for more various applications, especially in the...”

Reviewer 3 Report
This paper deals with the study on the effect of addition of anhydride (MAH) at different ratios together with graphene nanoplatelets (GnP) in poly(lactic acid)/modified natural rubber/polyaniline/GnP (PLA/m-NR/PANI/GnP) nanocomposites. For comparison purposes, two different techniques have been employed for preparation process: two-step technique and one-pot technique. The mechanical, morphological, and thermal properties of prepared nanocomposites were investigated. Although the manuscript is arranged roughly factually correct, the research described herein does not bring significant novelty over the previously reported works. After getting acquainted with the work in depth, the Reviewer came to the conclusion that the results of the studied samples are very similar, and the experiments did not bring spectacular results worthy for publication. Perhaps, more samples and tests would make this publication more attractive. In the introduction part, it should be included clear discussion of how this manuscript brings new information over papers that have already been published. The aforementioned evaluation pushed the Reviewer to suggest the rejection to this manuscript for publication in Polymers journal.
Additional comments:
1. The scientific novelty should be more strongly highlighted. What was done and achieved better over previous reports?
2. Lines 37-38: The authors mentioned that: “The environmentally friendly aliphatic polyester family, such as poly(lactic acid) (PLA), has been the best candidates” – Why? It should be explained.
3. The materials section needs to be organized with further details (e.g. particle size, specific surface area of graphene, PLA characteristic).
4. Lines 104-108: What temperature was selected for the PLA plasticization process and mixing process with NR?
5. How many specimens were tested in tensile experiment, which was the gas flow rate in the TG measurements, and how many scans were used for FTIR measurements? – The experimental procedures should be more detailed.
6. Line 136: “diffraction scanning calorimeter (DSC),” – should be differential.
7. Figure 2 – is it TGA curve from your experiments or from literature? It should be more specified and located in the discussion part or referenced to literature.
8. Line 208: “nanocompsoite” – should be nanocomposite.
9 Please revise references carefully according to the author's guidelines (there are some mistakes: e.g. reference 37 – year is not bolded and the journal name is not italic.
10. Section 3.3. : A table containing all wavenumbers and their description should be included in the manuscript. Importantly, the discussion related to FTIR results present in the manuscript seems to be strongly overinterpreted. In my opinion, the additional peaks suggested by the authors are not visible on the spectrum. Therefore, there is no reliable spectrocospective confirmation of the mechanism suggested by the authors in the study, which is also an important argument for rejecting the work.
Author Response
Response to Reviewer 3 Comments
Point 1: Perhaps, more samples and tests would make this publication more attractive.
Response 1: This study is aimed to focus on the investigation of the “compatibilization” effect in order to further improve the GnP/PANi dispersion and the tensile performance of the modified PLA matrix. The focus of this study is a preliminary-like study to determine the suitable grafting process method and MAH:GnP ratio based on the scientific assessment, this determination is important for us to proceed with the future functional characterization which is currently under on-going in our research work. However, we have intensively improved our manuscript by considering all the other comments and suggestions given by all the reviewers so that to make it scientifically meaningful for the readers.
Point 2: In the introduction part, it should be included clear discussion of how this manuscript brings new information over papers that have already been published.
Response 2: The introduction part has been intensively improved where a clear state-of-art discussion and the research gap/novelty of this current study are highlighted in Line 38-116.
Point 3: The scientific novelty should be more strongly highlighted. What was done and achieved better over previous reports?
Response 3: Thank you for the critical comment. A clearer state-of-art literature review with more latest references has been provided in the introduction part, together with the highlight of the scientific novelty of this current study (Line 38-116).
Point 4: Lines 37-38: The authors mentioned that: “The environmentally friendly aliphatic polyester family, such as poly(lactic acid) (PLA), has been the best candidates” – Why? It should be explained.
Response 4: This statement has been revised and adjusted as shown in Line 38-42.
Point 5: The materials section needs to be organized with further details (e.g. particle size, specific surface area of graphene, PLA characteristic).
Response 5: Further details of the raw materials have been provided in Table 1.
Point 6: Lines 104-108: What temperature was selected for the PLA plasticization process and mixing process with NR?
Response 6: The processing temperature for the PLA and LNR is 190°C as mentioned in the nanocomposite samples preparation (Line 156). The temperature (190°C) was fixed for the whole mixing process of the materials used in this study.
Point 7: How many specimens were tested in tensile experiment, which was the gas flow rate in the TG measurements, and how many scans were used for FTIR measurements? – The experimental procedures should be more detailed.
Response 7: The details of the characterisation part have been revised as requested as follow:
- The number of specimens for the tensile test is shown in Line 164-165: “Six specimens were tested to determine the average value.”
- The gas flow rate used in the TGA of 10 mL/min has been mentioned in the TGA characterisation part in Line 180: “…under an atmospheric nitrogen gas flow rate of 10 mL/min condition.”
-The number of scans for FTIR measurement has been incuded as shown in Line 174-175.
Point 8: Line 136: “diffraction scanning calorimeter (DSC),” – should be differential.
Response 8: We apologized for the mistake, the word “diffraction” has been changed to “differential” in Line 177.
Point 9: Figure 2 – is it TGA curve from your experiments or from literature? It should be more specified and located in the discussion part or referenced to literature.
Response 9: Figure 2 is drawn based on our own experimental results. We placed it in the characterization part so that it is easier for the readers to understand the integral procedure decomposition temperature (IPDT) calculation. In this sense, we have revised the sentence in Line 200 for better clarification.
Point 10: Line 208: “nanocompsoite” – should be nanocomposite.
Response 10: The spelling for nanocomposite has been revised in Line 249.
Point 11: Please revise references carefully according to the author's guidelines (there are some mistakes: e.g. reference 37 – year is not bolded and the journal name is not italic.
Response 11: The reference list has been revised accordingly.
Point 12: Section 3.3. : A table containing all wavenumbers and their description should be included in the manuscript. Importantly, the discussion related to FTIR results present in the manuscript seems to be strongly overinterpreted. In my opinion, the additional peaks suggested by the authors are not visible on the spectrum. Therefore, there is no reliable spectrocospective confirmation of the mechanism suggested by the authors in the study, which is also an important argument for rejecting the work.
Response 12: Thank you for the author’s comment on this. We have repeated our FTIR test and provided the new FTIR curve with a clear version of the additional peaks as shown in Figure 5.

Round 2
Reviewer 3 Report
The authors revised their manuscript according to my comments.
Author Response
Thank you.